

# Identification of biomarkers associated with clinical severity of chronic obstructive pulmonary disease

Jie Zhang[1,*], Changli Zhu[1,*], Hong Gao[1], Xun Liang[2], Xiaoqian Fan[3], Yulong Zheng[1], Song Chen[4] and Yufeng Wan[1]

[1] Department of Respiratory Diseases, The Affiliated Huai'an Hospital of Xuzhou Medical University, Huai'an, Jiangsu, China
[2] College of Nursing and Midwifery, Jiangsu College of Nursing, Huai'an, Jiangsu, China
[3] Department of Emergency Medicine, Suqian First Hospital, Suqian, Jiangsu, China
[4] Institute of Medicinal Biotechnology, Jiangsu College of Nursing, Huai'an, Jiangsu, China
* These authors contributed equally to this work.

Corresponding authors
Song Chen, biocs@163.com
Yufeng Wan, ggwanyufeng@163.com

## ABSTRACT

We sought to identify the biomarkers related to the clinical severity of stage I to stage IV chronic obstructive pulmonary disease (COPD). Gene expression profiles from the blood samples of COPD patients at each of the four stages were acquired from the Gene Expression Omnibus Database (GEO, accession number: GSE54837). Genes showing expression changes among the different stages were sorted by soft clustering. We performed functional enrichment, protein–protein interaction (PPI), and miRNA regulatory network analyses for the differentially expressed genes. The biomarkers associated with the clinical classification of COPD were selected from logistic regression models and the relationships between TLR2 and inflammatory factors were verified in clinical blood samples by qPCR and ELISA. Gene clusters demonstrating continuously rising or falling changes in expression (clusters 1, 2, and 7 and clusters 5, 6, and 8, respectively) from stage I to IV were defined as upregulated and downregulated genes, respectively, and further analyzed. The upregulated genes were enriched in functions associated with defense, inflammatory, or immune responses. The downregulated genes were associated with lymphocyte activation and cell activation. TLR2, HMOX1, and CD79A were hub proteins in the integrated network of PPI and miRNA regulatory networks. TLR2 and CD79A were significantly correlated with clinical classifications. TLR2 was closely associated with inflammatory responses during COPD progression. Functions associated with inflammatory and immune responses as well as lymphocyte activation may play important roles in the progression of COPD from stage I to IV. *TLR2* and *CD79A* may serve as potential biomarkers for the clinical severity of COPD. *TLR2* and *CD79A* may also serve as independent biomarkers in the clinical classification in COPD. TLR2 may play an important role in the inflammatory responses of COPD.

## INTRODUCTION

Chronic obstructive pulmonary disease (COPD) is a heterogeneous condition characterized by the progressive, irreversible limitation of a patient's airflow (*Agusti, 2014*; *Vestbo et al., 2013*). It is a leading cause of chronic morbidity and it has been predicted to be the third leading cause of mortality worldwide by 2030 (*Celli et al., 2015*; *Rodriguez-Roisin et al., 2017*). COPD is a progressive disease that typically worsens over time (*Rabe et al., 2013*). However, the molecular mechanisms underlying its progression are poorly understood (*Singh et al., 2018*). It has been reported that an early diagnosis and individualized treatments can reduce mortality and the socioeconomic burden caused by the disease. Damage to the airways and lungs may be delayed but not effectively reversed (*Pauwels et al., 2004*).

Spirometry is used to diagnose the severity of COPD by determining the degree of airflow limitation. Spirometry is based on the forced expiratory volume in one second (FEV1) expressed as a percentage of the predicted normal value for a person's gender, age, weight, and height (*Vestbo et al., 2013*). COPD is divided into four stages based on FEV1 according to the GOLD guidelines (*Vestbo et al., 2013*), and COPD exacerbations are classified as episodes of worsening symptoms from stage I to IV (*Wedzicha & Seemungal, 2007*). The treatment recommendations for European and American practitioners are based on FEV1 (*Qaseem et al., 2011*) but FEV1 does not directly reflect the systemic manifestation of symptoms in COPD patients (*Alotaibi & Ansari, 2016*). Several studies have demonstrated that there is a genetic component to COPD susceptibility (*Hersh et al., 2011*; *McCloskey et al., 2001*). Thus, the identification of these genetic markers may help to develop diagnostic and therapeutic targets for the treatment of COPD. Previous genome-wide association studies have identified several genetic loci related to COPD susceptibility (*Cho et al., 2011*; *Cho et al., 2014*). A recent study investigated the gene expression profiles among patients with frequent COPD exacerbations and identified three genes (*ARHGEF10*, *LAF4*, and *B3GNT*) as predictors of exacerbations (*Singh et al., 2014*). However, that study did not investigate whether those genes could be used as biomarkers for the clinical classification of COPD.

We downloaded the forementioned gene expression profile (GSE54837) (*Singh et al., 2014*) and investigated the genes showing changes in expression among different stages of COPD. We identified key genes using bioinformatics analyses and then further selected the biomarkers associated with the clinical classification of COPD using logistical regression models.

## MATERIALS AND METHODS

### Microarray dataset

We downloaded the gene expression profiles with accession number GSE54837 (*Singh et al., 2014*) from the Gene Expression Omnibus (GEO) database. These profiles included different clinical stage data, and were obtained based on the Affymetrix Human Genome U133 Plus 2.0 Array. We obtained 226 blood samples from 90 stage I patients, 58 stage II

patients, 55 stage III patients, and 13 stage IV patients. The age and sex of each patient is shown in Table S1.

## Data preprocessing

The original CEL data were preprocessed (background adjustment, quantile normalization, final summarization, and probe ID to gene symbol) using the RMA method (*Irizarry et al., 2003*) in the Affy package (*Gautier et al., 2004*). The expression values of the probes were averaged as the final gene expression value when the different probes corresponded to the same gene symbol. After the expression matrix of all genes in each sample was obtained, the gene expression values of each sample group with different clinical grades were averaged according to the clinical classification and the final gene expression differential matrix was obtained.

## Gene clustering analysis

The differential matrix of samples was subjected to noise-robust soft clustering using the fuzzy c-means algorithm in the Mfuzz package (*Futschik & Carlisle, 2005*; *Kumar & Futschik, 2007*). The Fuzzy C-Means (*Futschik & Carlisle, 2005*) clustering method was adopted for clustering analysis on the genes of samples using varying times and changes in expression levels. Multiple clustering results were obtained, and the results were divided according to the trends into two parts: rising and falling. The parameters used were minSTD = 0.1, acore = 0.5, and cOptimal = 10.

## Functional enrichment analyses

The gene sets with significant rising or falling trends were examined using the Gene Ontology (GO) and Kyoto Encyclopedia of Genes and Genomes (KEGG) pathway enrichment analyses using the Database for Annotation, Visualization, and Integrated Discovery (DAVID) (*Huang, Sherman & Lempicki, 2008*). Enrichment results were identified in biological processes (BP), cellular components (CC), molecular functions (MF), and pathways.

## Protein–Protein Interaction (PPI) analysis and miRNA-target gene regulatory relation analysis

PPI pairs were predicted based on the genes with rising and falling trends using the Search Tool for the Retrieval of Interacting Genes (STRING) (*Mering et al., 2003*). The required confidence of (combined score) > 0.4 was set as the threshold value for PPI relation prediction.

The regulatory miRNAs of genes with PPI relations were predicted using Webgestalt (*Zhang, Kirov & Snoddy, 2005*) based on the hypergeometric test with a threshold of enriched gene count ≥ 3. The $p$ value was adjusted using the Benjamini–Hochberg (BH) method (*Benjamini & Hochberg, 1995*), and the adjusted $p$ value < 0.05 was set as the threshold.
## Integration of PPI and miRNA-target gene regulatory relations

The PPI pairs and miRNA-target gene regulatory relation pairs were integrated to construct a network using Cytoscape (*Shannon et al., 2003*). The degree of connectivity in the network was analyzed and the hub protein (the node with a high degree of connectivity) was selected.

## Influencing factors of the clinical grade of COPD

We performed logistical regression analysis after obtaining the hub proteins in the network based on the age ($\leq 60 = 0$, $>60 = 1$), gender (female $= 0$, male $= 1$), and the gene corresponding to the hub protein as potential factors influencing the clinical grade to further determine whether the expression of the genes could be used as biomarkers of clinical grade. Univariate logistic regression analysis was conducted to preliminarily screen the factors with *p* value $< 0.05$. Multivariate logistic regression analysis was then conducted to further select the factors with *p* value $< 0.05$.

## Clinical data collections and detections

Investigators obtained written informed consent before enrolling participants in the clinical trial. We selected patients who underwent pulmonary function testing at the respiratory function laboratory of the Affiliated Huai'an Hospital of Xuzhou Medical University between January 1, 2019 and May 31, 2019. Patients with complications including asthma, pulmonary interstitial fibrosis, bronchial dilatation, other pulmonary diseases, or malignant tumors were excluded. COPD was staged according to the GOLD guidelines: GOLD I, FEV1 $\geq 80\%$ predicted; GOLD II, FEV1 $< 80\%$ and $\geq 50\%$ predicted; GOLD III, FEV1 $< 50\%$ and $\geq 30\%$ predicted and GOLD IV, FEV1 $< 30\%$ predicted. Patients selected for the study were divided into the following four groups: no related diseases non-smoker, no related diseases smoker, COPD stage I-II, and COPD III-IV. Peripheral venous blood samples were collected from all subjects and placed in anticoagulation centrifuge tubes. Mononuclear cells were separated by density gradient centrifugation. CD14$^+$ immunomagnetic beads (Miltenyi, Germany) were sorted by the automatic magnetic bead sorter. TLR2 gene expression in the monocytes was detected using real-time quantitative PCR according to the instructions of the Real-time PCR kit (Takara, Tokyo, Japan). The amplification specificity was confirmed by melting curves and fluorescence was determined at 60 °C. PCR conditions were as follows: predegeneration at 95 °C for 30 s, followed by 40 cycles for 5 s at 95 °C and 31 s at 60 °C. qRT-PCR reactions were performed in a total volume of 20 µL. The primer sequences for TLR2 were as follows: F: 5′-TCGGAGTTCTCCCAG TTCTCT-3′, R: 5′-TCCAGTGCTTCAACCCACAA -3′. The internal reference was β-actin F: 5′-CCTGGCACCCAGCACAAT-3′, R:5′-GGGCCGGACTCGTCATAC3′. Additional blood samples were taken and centrifuged at 3,000 r/min for 10 min, after which the serum was separated and stored in a refrigerator at −80 °C for testing. All experiments were performed in triplicate. Relative quantitative expression levels were calculated using the $2^{-\Delta\Delta Ct}$ method. The levels of IL-6, IL-8, TNF-α, and IFN-γ in the serum samples were detected and compared by an ELISA Kit (Sangon Biotech, Shanghai, China). All experiments were approved by the Ethics Committee of the Affiliated Huai'an Hospital of Xuzhou Medical University. The clinical characteristics of all the subjects are shown in Table S2.

## Statistical analysis

We used SPSS 19.0 and Graphpad Prism 7.0 for statistical analysis. The expression levels of TLR2 and inflammatory factors were analyzed using a one-way ANOVA, followed by the least significant difference multiple comparison post-hoc test, when appropriate. We analyzed the correlation between TLR2 expression levels and inflammatory factor levels in all detected samples using Pearson Correlation Analysis. All statistical analyses were conducted with a significance level of $\alpha = 0.05$ ($P < 0.05$).

# RESULTS

## Gene clustering analysis

Opposing trends were observed among clusters 1, 2, and 7 and clusters 5, 6, and 8 as the clinical stage progressed from stage I to stage IV (Fig. 1). Clusters 5, 6, and 8 showed significantly increased expression, while clusters 1, 2, and 7 displayed significantly decreased expressions from stage I to IV. The changes in expression in clusters 3, 4, 9, and 10 were disordered. We selected clusters 1, 2, 5, 6, 7, and 8 for further analysis. A total of 78, 42, and 77 genes were included in clusters 5, 6, and 8, respectively, and were defined as upregulated genes. 59, 44, and 56 genes were included in clusters 1, 2, and 7, respectively, and were defined as downregulated genes.

## Functional enrichment analyses

The upregulated genes were significantly related to GO functions including defense, inflammatory, wounding, and immune responses, the positive regulation of biosynthetic processes, system lupus erythematosus pathways, the toll-like receptor signaling pathway, and cytokine-cytokine receptor interactions (Table 1). The downregulated genes were involved in GO functions including lymphocyte, cell, and leukocyte activation, the external side of the plasma membrane, type I diabetes mellitus pathway signaling, Jak-STAT signaling, and T cell receptor signaling (Table 2).

## PPI and miRNA-target gene regulatory relations prediction

100 pairs were predicted from upregulated genes and 40 PPI pairs were predicted from downregulated genes using STRING. 64 and 92 miRNA-target gene regulatory relation pairs were predicted from upregulated and downregulated genes, respectively (Table 3).

## Network integration analysis

Two integrated networks were constructed based on the obtained PPI and miRNA-target gene regulatory relation pairs (Fig. 2). A constructed network of upregulated genes displayed the following five nodes with degrees ≥ 6: toll like receptor 2 (TLR2), matrix metallopeptidase 9 (MMP9), heme oxygenase 1 (HMOX1), and C-C motif chemokine receptor 1 (CCR1) (Fig. 2A). In the network of downregulated genes, the CD79a molecule (CD79A) and phospholipase C gamma 1 (PLCG1) had degrees ≥ 6 and were considered to be hub nodes (Fig. 2B).

## Influencing factors of the clinical grade of COPD

Univariate logistic regression analysis results revealed that the factors of age, *TLR2*, *MMP9*, *CCR1*, *CD79A*, and *PLCG1* were significantly associated with clinical classification (Table 4).

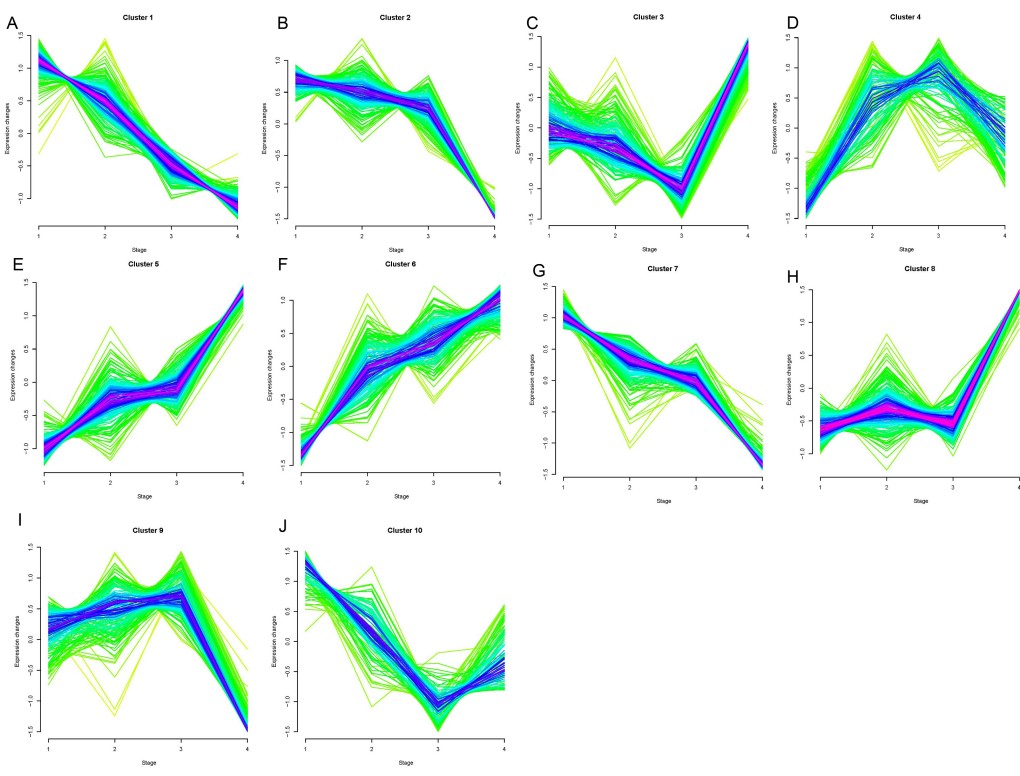

**Figure 1  Results of soft clustering for genes.** Changes in colors (red-blue–green) indicate the coincidence degree of a gene change with the central variation of the cluster. Red indicates a high degree and green indicates a low degree of coincidence.Cluster1-10 (A–J).

**Table 1  Upregulated genes with enriched functions of Gene Ontology (GO) and the Kyoto Encyclopedia of Genes and Genomes (KEGG).**

| Category | Term | Description | Count | P value |
|----------|------|-------------|-------|---------|
| BP | GO:0006952 | defense response | 22 | 3.84E−07 |
| BP | GO:0006954 | inflammatory response | 16 | 5.18E−07 |
| BP | GO:0009611 | response to wounding | 20 | 7.08E−07 |
| BP | GO:0006955 | immune response | 22 | 2.45E−06 |
| CC | GO:0009891 | positive regulation of biosynthetic process | 35 | 0.002989929 |
| CC | GO:0051173 | positive regulation of nitrogen compound metabolic process | 22 | 0.004929606 |
| CC | GO:0031328 | positive regulation of cellular biosynthetic process | 22 | 0.006361208 |
| CC | GO:0045935 | positive regulation of nucleobase, nucleoside, nucleotide, and nucleic acid metabolic process | 51 | 0.006484061 |
| KEGG | hsa05322 | Systemic lupus erythematosus | 8 | 4.11E−04 |
| KEGG | hsa04620 | Toll-like receptor signaling pathway | 6 | 0.012939 |
| KEGG | hsa04060 | Cytokine-cytokine receptor interaction | 9 | 0.028464 |

**Notes.**
BP, biological process; CC, cellular component.
**Table 2** Downregulated genes with enriched functions of Gene Ontology (GO) and the Kyoto Encyclopedia of Genes and Genomes (KEGG).

| Category | Term | Description | Count | P value |
|---|---|---|---|---|
| BP | GO:0046649 | lymphocyte activation | 11 | 1.89E−06 |
| BP | GO:0001775 | cell activation | 12 | 7.79E−06 |
| BP | GO:0045321 | leukocyte activation | 11 | 1.08E−05 |
| BP | GO:0048534 | lymphoid organ development | 10 | 1.19E−04 |
| BP | GO:0002520 | immune system development | 10 | 1.87E−04 |
| BP | GO:0030097 | hemopoiesis | 9 | 3.33E−04 |
| BP | GO:0030098 | lymphocyte differentiation | 6 | 9.42E−04 |
| BP | GO:0042110 | T cell activation | 6 | 0.002303352 |
| BP | GO:0002521 | leukocyte differentiation | 6 | 0.0027276 |
| BP | GO:0042113 | B cell activation | 4 | 0.018182557 |
| BP | GO:0030217 | T cell differentiation | 3 | 0.081968615 |
| CC | GO:0009897 | external side of plasma membrane | 4 | 0.120963049 |
| CC | GO:0009986 | cell surface | 5 | 0.235650749 |
| CC | GO:0044459 | plasma membrane part | 24 | 0.033231996 |
| CC | GO:0031226 | intrinsic to plasma membrane | 14 | 0.088759042 |
| CC | GO:0005886 | plasma membrane | 34 | 0.099440061 |
| CC | GO:0005887 | integral to plasma membrane | 13 | 0.136364905 |
| KEGG | hsa04940 | Type I diabetes mellitus | 4 | 0.001194 |
| KEGG | hsa04630 | Jak-STAT signaling pathway | 5 | 0.007356 |
| KEGG | hsa05330 | Allograft rejection | 3 | 0.014236 |
| KEGG | hsa05332 | Graft-versus-host disease | 3 | 0.016588 |
| KEGG | hsa04660 | T cell receptor signaling pathway | 4 | 0.016985 |

Notes.
BP, biological process; CC, cellular component.

These factors were further analyzed by multivariate logistic regression analysis and *TLR2* was found to be significantly positively correlated and *CD79A* was significantly negatively correlated with clinical classification, suggesting that they could be used as independent biomarkers of clinical classification (Table 5).

## The relationships between TLR2 and inflammatory factors in COPD progression

TLR2 can trigger signal transduction and lead to the release of inflammatory mediators. COPD is defined as a chronic inflammatory lung disease. We chose IL-6, IL-8, TNF-α, and IFN-γ to assess the severity of COPD as they are known inflammatory factors in COPD progression. We found that the levels of IL-6, IL-8, TNF-α, and IFN-γ in the smoker and COPD III-IV groups were significant higher than those in the non-smoker and COPD I-II groups, and TLR2 in mononuclear cells of the peripheral blood was significantly correlated with the clinical classification of the patients (Figs. 3A–3E). We also found that TLR2 and the expressions of inflammatory markers were more significant in the GOLDIII/VI group than those in the healthy smoker group. We used the ROC curves to identify the sensitivity and specificity of the expression levels. The results showed that the five markers, IL-6, IL-8, TNF-α, IFN-γ and TLR2, have diagnostic value for COPD. The areas under the curve (AUC)

**Table 3  Regulatory microRNAs of gene sets.**

| Sequence | microRNA | Count |
|---|---|---|
| Hsa-TGGTGCT | miR-29a, miR-29b, miR-29c | 6 |
| Hsa-AAGGGAT | miR-188 | 3 |
| Hsa-GTGCCTT | miR-506 | 7 |
| Hsa-ACACTAC | miR-142-3p | 3 |
| Hsa-TGCACTG | miR-148a, miR-152, miR-148b | 4 |
| Hsa-ACTGAAA | miR-30a-3p, miR-30e—3p | 3 |
| Hsa-GGGACCA | miR-133a, miR-133b | 3 |
| Hsa-CTTTGCA | miR-527 | 3 |
| Hsa-CATTTCA | miR-203 | 3 |
| Hsa-TTTGTAG | miR-520d | 3 |
| Hsa-TAGCTTT | miR-9 | 4 |
| Hsa-AAGCACA | miR-218 | 4 |
| Hsa-TGGTGCT | miR-29a, miR-29b, miR-29c | 4 |
| Hsa-GCACTTT | miR-17-5p, miR-20a, miR-106a, miR-106b, miR-20b, miR-519d | 4 |
| Hsa-AAGCCAT | miR-135a, miR-135b | 3 |
| Hsa-TGCACTT | miR-519c, miR-519b, miR-519a | 3 |
| Hsa-CTTTGTA, | miR-524 | 3 |
| Hsa-TGAATGT | miR-181a, miR-181b, miR-181c, miR-181d | 3 |
| Hsa-TGCTGCT | miR-15a, miR-16, miR-15b, miR-195, miR-424, miR-497 | 3 |

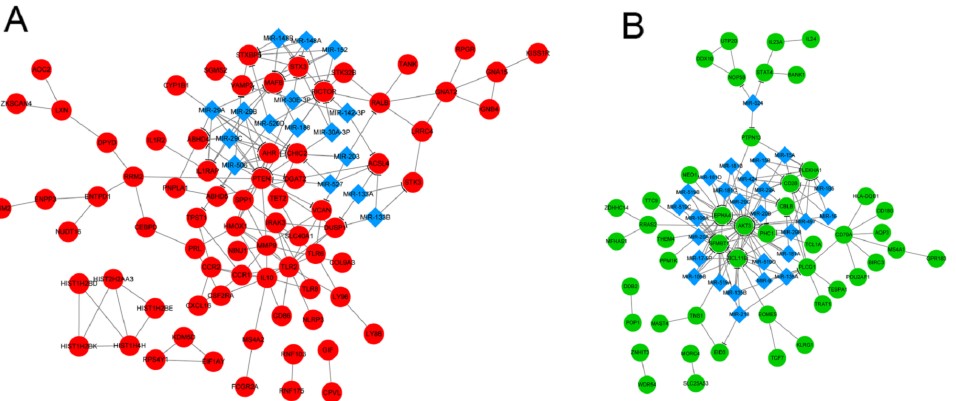

**Figure 2  The integrated network of protein–protein interactions and miRNA-target gene regulatory relations for upregulated genes (A) and downregulated genes (B).** Red circles indicate upregulated genes, green circles indicate downregulated genes, and blue diamonds indicate miRNA.

were 0.7782, 0.8857, 0.9318, 0.9276 and 0.7647, respectively ($P < 0.0001$, Figs. 3F–3J). The correlations between the TLR2 expression and inflammatory factor levels was analyzed by Pearson correlation analysis. The expression levels of TLR2 were correlated with the serum levels of IL-6 (standardized regression coefficient = 0.5594, $P < 0.0001$, Fig. 3K), IL-8 (standardized regression coefficient = 0.6498, $P < 0.0001$, Fig. 3L), TNF-α (standardized

**Table 4  Results of univariate logistic regression analysis.**

| Factor | Coefficient | P value |
|---|---|---|
| Age | 0.2796 | 0.0389[*] |
| Gender | 0.1545 | 0.2383 |
| TLR2 | 0.9265 | 2.1068E−07[***] |
| IL-10 | 0.3250 | 0.1524 |
| MMP9 | 0.2702 | 0.0059[**] |
| HMOX1 | 0.3163 | 0.0713 |
| CCR1 | 0.3503 | 0.0298[*] |
| CD79A | −0.3443 | 0.0002[***] |
| PLCG1 | −0.7284 | 0.0037[**] |

Notes.
Factor, factors that may affect clinical grading; coefficient, regression coefficient ( > 0, positive 3 correlation; < 0, negative correlation).

**Table 5  Results of multivariate logistic regression analysis.**

| Factor | Coefficients | P value |
|---|---|---|
| Age | 0.1069 | 0.4220 |
| TLR2 | 0.6894 | 0.0027[**] |
| MMP9 | 0.0374 | 0.7310 |
| CCR1 | 0.0282 | 0.8695 |
| CD79A | −0.1930 | 0.0450[*] |
| PLCG1 | −0.1267 | 0.6488 |

Notes.
Factor, factors that may affect clinical grading; coefficient, regression coefficient (> 0, positive 3 correlation; < 0, negative correlation).

regression coefficient = 0.5126, $P < 0.0001$, Fig. 3M), and IFN-$\gamma$ (standardized regression coefficient = 0.4973, $P < 0.0001$, Fig. 3N).

## DISCUSSION

COPD is a growing global health problem. As the population ages, life expectancy among the elderly is expected to rise in relation to the number of smokers and the prevalence of COPD. Researchers are working to identify and diagnose COPD using easily available biomarkers to facilitate the early detection of fluid build-up in COPD and improve diagnostic accuracy. Previous studies focused on finding new biomarkers in the blood for the early diagnosis of COPD and many of these potential diagnostic markers need to be verified in additional studies.

We studied gene clusters with continuously rising (clusters 5, 6, and 8) or falling (clusters 1, 2, and 7) trends in expression changes from stage I to stage IV. The upregulated genes, such as *TLR2*, and *HMOX1*, were significantly enriched in functions associated with defense, inflammatory, or immune responses, while the downregulated genes, such as *CD79A*, were associated with the activation of lymphocytes, cells, and leukocytes. TLR2, HMOX1, and CD79A were hub proteins in the integrated network. Logistic regression

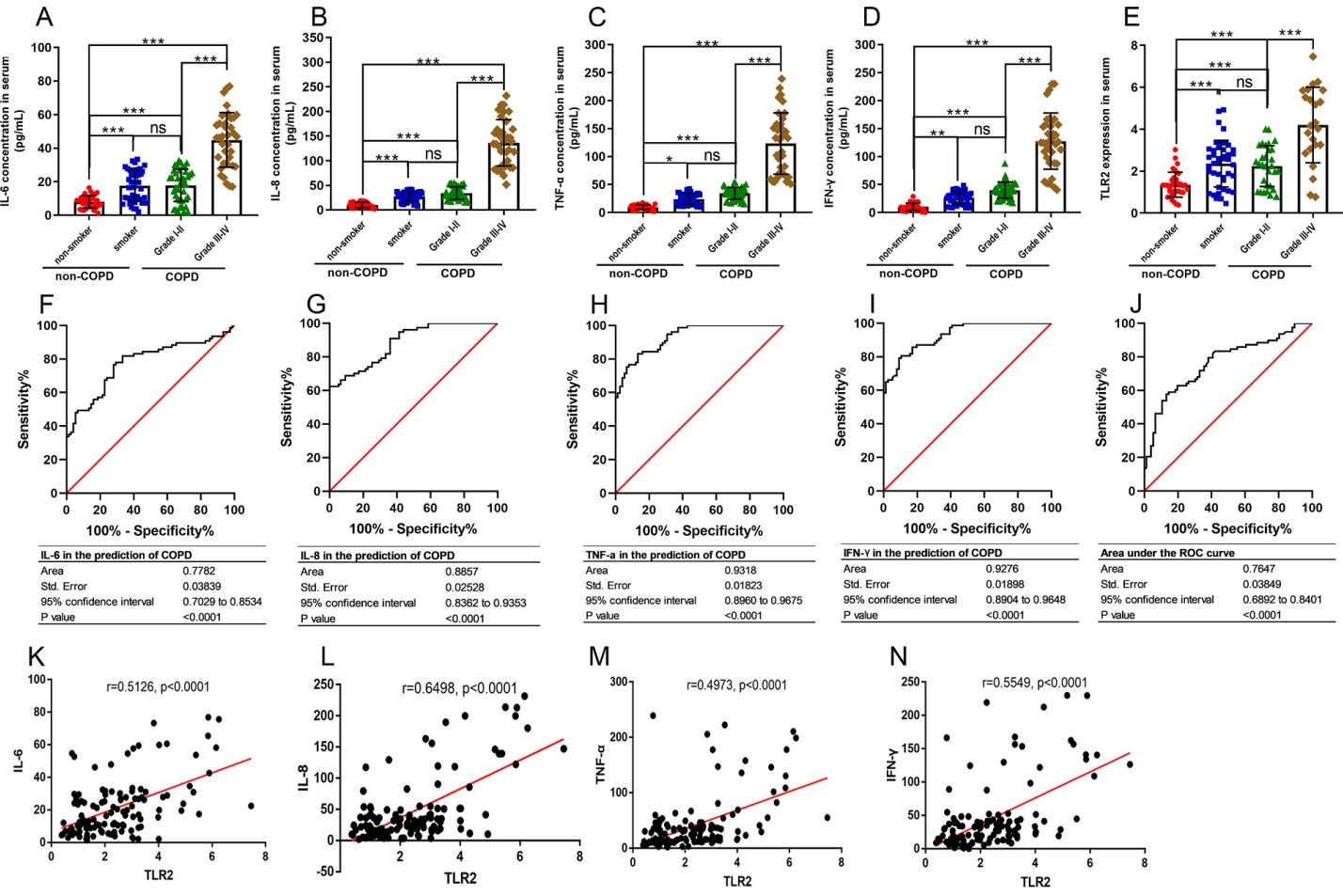

**Figure 3** **The relationships between TLR2 and inflammatory factors in COPD progression.** The levels of IL-6 (A), IL-8 (B), TNF-α (C), and IFN-γ (D) in the serum and TLR2 in the peripheral blood mononuclear cells of the patients (E). The sensitivity and specificity of IL-6 (F), IL-8 (G), TNF-α (H), IFN-γ (I) and TLR2 (J) in the prediction of COPD . The TLR2 expression levels were correlated with the serum levels of IL-6 (K), IL-8 (L), TNF-α (M), and IFN-γ (N).

analysis showed that *TLR2* was positively correlated and *CD79A* was negatively correlated with clinical classification.

Our results showed that the upregulated genes, such as *TLR2*, and *HMOX1*, were significantly associated with the inflammatory and immune responses. Exposure to inhaled pollutants can result in chronic airway inflammation in COPD by activating structural and inflammatory cells within the lungs (*Rovina, Koutsoukou & Koulouris, 2013*). COPD exacerbation is defined as an acute worsening of the symptoms that are implicated in increased systemic and airway inflammation (*Wedzicha & Seemungal, 2007*). The inflammatory and immune responses are known to play a critical role in COPD. Dave Singh, et al. (2014b) found that systemic immune function is associated with the exacerbation of COPD. Our study further suggested the important role of inflammatory and immune responses in the progression of COPD from stage I to stage IV. IL-6, IL-8, TNF-α and IFN-γ are important inflammatory mediators during the progression of COPD and IL-6

participates in the inflammatory response. It can further aggravate the oxidative stress and inflammatory responses by promoting the proliferation and maturation of T lymphocytes and B lymphocytes. IL-8 is an inflammatory mediator that promotes the activation of neutrophils and the release of various enzymes that damage the bronchi. Excessive levels of IL-8 can aggravate the inflammatory response and destroy the lung tissue (*Farahi et al., 2017*; *Nakamoto et al., 2019*). By acting on neutrophils, TNF-α enhances the expression of leukocyte adhesion molecules and activates other factors involved in the inflammatory response. Higher levels of TNF-α and IFN-γ in COPD also indicate increased inflammation (*Xu, Li & Sun, 2019*; *Zhang et al., 2016*). Our findings about these inflammatory factors in COPD were consistent with previous studies.

TLR2 was among the top two hub proteins in the network. TLR2 is a member of the toll-like receptor family, which plays a fundamental role in the activation of innate immunity (*Da Silva et al., 2008*). *Haw et al. (2018)* found that TLR2 mRNA expression was increased in the epithelium and parenchyma of the mouse airway when chronically exposed to cigarette smoke. These results were replicated in human COPD patients. Our results showed that increased *TLR2* was an independent clinical classification biomarker in COPD. The results of some studies do not support this research. A previous study reported that the alveolar macrophages in smokers and COPD patients presented an equally decreased surface expression of TLR2 compared to non-smokers (*Droemann et al., 2005*). The role of TLR2 in the pathogenesis of COPD is also controversial in previous literature (*Freeman et al., 2013*; *Haw et al., 2018*; *Simpson et al., 2013*; *Von Scheele et al., 2011*). These conflicting results may be due to differences in experimental methods (e.g., peripheral blood monocytes vs. macrophages) or cohorts of patients with varying medical backgrounds.

We found that the downregulated genes were involved in functions associated with lymphocyte activation, which may suggest that lymphocyte activation plays a role in COPD development. COPD has been thought to possess an autoimmune component (*Agusti et al., 2003*; *Lee et al., 2007*), but the antigenic stimulation responsible for lymphocyte activation is unclear. Previous studies demonstrated that the numbers of B lymphocytes and CD8+ T cells are increased in the airways of COPD patients compared with healthy controls (*Gosman et al., 2006*; *Saetta et al., 1999*). The number of lymphocytes has been shown to increase with the severity of the disease (*Hogg et al., 2004*). We found that the hub protein, CD79A, was enriched with lymphocyte activation. CD79A has been demonstrated to play diverse roles in the development and function of B lymphocytes (*Pelanda et al., 2002*) and may be downregulated in severe COPD when compared with controls (*Cockayne et al., 2011*). Our results showed that CD79A was negatively correlated with the clinical classification of COPD. Thus, lymphocyte activation and *CD79A* may be associated with the clinical severity of COPD.

Our study is limited because we analyzed the differentially expressed genes in COPD patients using a single set of GEO data. The sample capacity was small and more GEO data should be used in further studies. Secondly, by extracting the total RNA content in blood monocytes, we found that the Ct value was relatively large, indicating that the expression of TLR2 in blood was lower, and the micro expression levels may have caused experimental

errors. We performed each test at least three times to improve the repeatability of the results. Finally, we only studied the expression of TLR2 in the clinical population but its specific mechanism and inconsistency in previous studies was well explained and requires additional experimentation. Our results show that the inflammatory and immune responses, as well as lymphocyte activation, may play important roles in COPD development from stage I to stage IV. *TLR2*, and *CD79A* may serve as potential biomarkers in the exacerbation of COPD, and *TLR2* and *CD79A* may also serve as independent biomarkers for the clinical classification of COPD.

## ACKNOWLEDGEMENTS

We wish to thank Mr. Lizhang Xun for assisting with the collection of clinical blood samples and Prof. Song Chen for their assistance with the experiments.

### Funding

This study was supported by grants from Jiangsu Province's Key Talents Training Program of Youth Medicine (QNRC2016426). The funders had no role in study design, data collection and analysis, decision to publish, or preparation of the manuscript.

### Grant Disclosures

The following grant information was disclosed by the authors:
Jiangsu Province's Key Talents Training Program of Youth Medicine: QNRC2016426.

### Competing Interests

The authors declare there are no competing interests.

### Author Contributions

- Jie Zhang conceived and designed the experiments, performed the experiments, analyzed the data, prepared figures and/or tables, and approved the final draft.
- Changli Zhu conceived and designed the experiments, performed the experiments, prepared figures and/or tables, and approved the final draft
- Hong Gao, Xun Liang and Xiaoqian Fan performed the experiments, prepared figures and/or tables, and approved the final draft
- Yulong Zheng, Song Chen and Yufeng Wan conceived and designed the experiments, authored or reviewed drafts of the paper, and approved the final draft

### Data Availability

   The raw measurements are available in the Supplemental Files.

### Supplemental Information

Supplemental information for this article can be found online at http://dx.doi.org/10.7717/peerj.10513#supplemental-information.

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
