# Peer review of "Identification of biomarkers associated with clinical severity of chronic obstructive pulmonary disease"

_PeerJ, doi:10.7717/peerj.10513_

## Round 0.1 · original submission · Major Revisions

Both the reviewers have raised multiple major questions. Please address them point by point in the revised manuscript. Also, Please include any missing data highlighted by the reviewer ( Cluster 7-10) .

Reviewer 1 ·

Basic reporting

The present work is to the greatest extant comprehensible written and well structured.

The literature, especially in the discusion part should be extended:
In the discussion the findings should be set in relationship to other studies investigating biomarkers for COPD severity and exacerbation. For example:
a. Work of Singh et al. Source Paper – Plos ONE 2014
b. Rogers, 2019, PlosONE
c. Ect.

The article is well structured, however there are some missing parts:
Gene clustering analysis (Figure 1): Cluster 7-10 are missing in the figure.

Dataset of the GEO Omnibus: please add link in method section.

Please find further details in general comments.

Experimental design

The methodical setup to analyze the data sets sounds solid. However, a graphical analysis pipeline should be added to improve the clarity of work flow.

Precise description of patient cohorts are missing (the own and GEO database)

A link to the GEO Databaseset should be added

Please find further details in general comments.

Validity of the findings

Conclusion should be edited: tha analysis demonstrate markers for clinical severity of COPD but not exacerbation.

The discussion part should contain the discussion with other works analyzing biomarkers for COPD.

The final analysis of TLR2 expression in own clinical samples is not completely meet the demands that TLR2 is a marker for GOLD III/IV, since no differences to healthy smokers are observeable.

Please find further details in general comments.

Additional comments

Data analyzed in the present work based on a microarray data set (GSE54837) published by Singh et al. 2014 (PLOS ONE). In this work Singh et. al analyzed genes associated with COPD exacerbation in 138 sputum and blood samples.
In the present study Zhang et. al. used the microarray data set (received from the GEO database) to analyzes differential regulated genes in blood samples of 226 patients which were staged based on their lung function parameters to GOLD I (n=90), II (n=58), III (n=55) and IV (n=13). Using data preprocessing and cluster analyses they identified different set of clusters up or down regulated from GOLD I to GOLD IV. Using functional enrichment analyses they could demonstrate, that up-regulated genes were associated with defense, inflammation or immune responses, whereas down-regulated genes were associated with cell activation. Further network analyses based on data received from protein-protein interaction analyses and miRNA-target gene regulatory relation analysis revealed five nodes associated with upregulated genes (TLR2, IL-10, MMP9, HMOX1 and CCR1) and two nodes associated with down-regulated genes (CD79A and PLCG1). Analyzing blood samples from an own clinical cohort the group demonstrated that TLR-2 mRNA expression levels are increased on PBMC of GOLD III/IV COPD patients in comparison to GOLD I/II patients and that TLR-2 expression correlated with serum levels of inflammatory mediators (IL-6, IL-8, TNF-a and IFNy).

The present work is to the greatest extent comprehensible written and well structured.

There are several points which should be clarified before considering the manuscript for publication:

1. The work want to identify biomarkers associated with COPD exacerbation, an objective also the publication which was source of the data set was aiming for.
However, different from the original article the present manuscript did not differentiate patients based on exacerbations per year but on degree of disease severity derived from the lung function parameter FEV1 and staged the patient from GOLD I to GOLD IV.
So that the present markers are associated with the severity of the disease (clinical grade) but not with the susceptibility to develop exacerbations.

2. Beside the description of the own patient cohort also the cohort of the original study should be described and at least classification of patients in subgroups should be explained.

3. A detailed information on the own patient cohort is missing. Information like number of subjects, disease state, age, sex ect. should be added.

4. TLR-2 was chosen to demonstrate the clinical importance as a possible biomarker in an own clinical cohort. Unfortunately, there seem to be no differences in the expression between GOLD IV patients and healthy smokers.
a. Give the other top up- or down regulated marker better results?
b. How were the correlations of TLR-2 to the inflammatory cytokines performed? Were all samples taken or just within the subgroups.

5. How is the sensitivity and specificity of the identified marker?

6. Can you confirm the results with other free data sets?
a. Especially the subject number in the GOLD IV group is quite low and should be increased.

7. In the discussion the findings should be set in relationship to other studies investigating
biomarkers for COPD severity and exacerbation. For example:
a. Work of Singh et al. Source Paper – Plos ONE 2014
b. Rogers, 2019, PlosONE
c. Ect.

8. Discussion:
a. Please edit Line 271-282, this part is quite confusing because lower expression in COPD of TLR2 and IL-10 in other works are described, you demonstrate higher levels but do not discuss the differences.
b. 283-294: You describe that lymphocytes like CD8 and B cells are increased in COPD and that the number of lymphocytes are associated with COPD, however, CD79A was downregulated. Please discuss this observation.

9. Gene clustering analysis (Figure 1): Cluster 7-10 are missing in the figure.

10. The methodical setup to analyze the data sets sounds solid. However, a graphical analysis pipeline should be added to improve the clarity of work flow.

11. Dataset of the GEO Omnibus: please add link in method section.

12. Advantages and disadvantages of the present study should be described. Please explain the advantage of the biomarkers to identify the clinical phenotype in comparison to the standard method, the assessment of lung function and number of exacerbations.

13. There are some minor typos which should be corrected. For example: line 114 molecular functions (MF), and pathways..

·

Basic reporting

no comment

Experimental design

no comment

Validity of the findings

no comment

Additional comments

The manuscript entitled “Identification of biomarkers associated with chronic obstructive pulmonary disease exacerbation from mild to severe” by Zhang et al. studies gene expression profiles in blood samples from COPD patients in order to identify biomarkers associated with COPD. The authors used Gene Expression Omnibus Database to obtain gene expression profiles of 226 blood samples. Thereafter, they performed gene clustering analysis followed by pathway enrichment analysis. Based on the obtained results, the miRNA of genes with PPI relations were predicted. The identified biomarkers were TLR2 and CD79A where the relationship between TLR2 and the inflammatory markers from the clinical samples of COPD patients was verified.

1. The manuscript is well written. The analyses are appropriate and statistics are well used.
2. The abstract has to be rewritten according to the flow of the manuscript (line 30, 'finally....' should be at the end of the abstract).
3. Experimental design is optimal. Nevertheless, in the description for Fig 1 (line 177-178 and 180), authors mention 10 clusters that were chosen for further analysis. However, there are only 6 clusters in the figure; clusters 7, 8, 9 and 10 are missing.
4. Please provide the sequence of the primer used.
5. The authors suggest that TLR2 and CD79A can be used as independent biomarkers for clinical COPD classification, therefore, it would be interesting to see the relationship between CD79A and inflammatory factors as well (the authors have only provided the PCR and ELISA results for TLR2).
6. Including more speculations about significance of the findings in the discussion section will add more value to the text.

---

## Round 0.2 · Major Revisions

Reviewer 1 has raised several questions which have to be addressed before moving forward. Please address them carefully and resubmit the manuscript.

Reviewer 1 ·

Basic reporting

Pleas find specific information in General Comments

Please check orthography, and typos, there are still sentences and sections which are hard to understand.

Figure legends should be extendend.

Some methodical descriptions should be extended.

Results have to be added to improve the confirmation of the hypothesis.

Experimental design

Some methodical descriptions should be extended.

Validity of the findings

I have still the feeling that the discussion part is incomplete. Here own results should be set in relationship with findings of others.

What is already known about biomarkers of COPD severity and exacerbation, how fit the new detected biomarkers in the present picture. Where are similarities and differences.

Still a critical dispute with the own work are missing. What are downsides and benefits of analyzing potential biomarkers via micro assay data of blood.

Please state helpfulness of biomarkers analysis beside classical classification of the disease state.

Additional comments

In the present work a data set generated by Singh et al. 2014 (Plos One) was used to identify biomarkers associated with the severity of COPD.
Here, TLR-2, IL-10 and CD79A were highlighted as potential serum biomarkers for COPD severity.
The group tried to confirm differentiated expression of TLR-2 in sera of patients and healthy volunteers of an own COPD cohort.
Here they demonstrate that TLR-2 expression is increased in GOLD III-IV COPD patients. And that TLR-2 expression is positive correlated with inflammation markers like IL-6, IL-8, TNFa and IFNy.
Unfortunately, several improvement suggestions of the reviewers in the first phase of the process were not or incompletely answered.
Here, important information is still missing:

1. Reviewer 1: Comment 4:

TLR-2 was chosen to demonstrate the clinical importance as a possible biomarker in an own clinical cohort. Unfortunately, there seem to be no differences in the expression between GOLD IV patients and healthy smokers.
a. Give the other top up- or down regulated marker better results?
b. How were the correlations of TLR-2 to the inflammatory cytokines performed? Were all samples taken or just within the subgroups.
Response: Thank you for your constructive comments. We have added some disscussions about the results. And the pearson correlation analysis was used to investigate the correlations of TLR-2 to the inflammatory cytokines. We have added it in the methods part.

Reviewer Response:
There are still no information given if the TLR2 expression is significantly different between GOLDIII/VI and healthy smokers. There is also no statistically comparison of the other inflammatory marker between the both groups.
The group did not answer the question if correlations were performed with all samples of all patients and healthy volunteers or within each subgroup (healthy, healthy smoker, COPD I/II, COPD III/IV).

2. Reviewer 2: Comment 5.
The authors suggest that TLR2 and CD79A can be used as independent biomarkers for clinical COPD classification, therefore, it would be interesting to see the relationship between CD79A and inflammatory factors as well (the authors have only provided the PCR and ELISA results for TLR2).
Response: Thank you very much for your comments. The hub protein CD79A was enriched in the function of lymphocyte activation. CD79A has been demonstrated to play diverse roles in the development and function of B lymphocytes. The relationship between CA79A and COPD is not well understood until now. Importantly, our results showed that CD79A was negatively correlated with the clinical classification of COPD. However, the relationship between immunity and inflammation is complex. In the whole immune system, CD79A is not a key marker at the core position, and the simple expression level of CD79A is hard to explain anything. Based on this, we did not study the expression relationship between CD79A and inflammatory factors furtherly.

Reviewer Response:
The authors answer that it is not meaning full to relate CD79A expression to inflammatory markers.
Since CD79A is one of their top Biomarkers I would recommend that comparable to TLR-2 the negative association of this marker and also IL-10 should be demonstrated in their cohort. These results will strengthen the importance of their observations.


3. Reviewer 1: Comment 5.
The methodical setup to analyze the data sets sounds solid. However, a graphical analysis pipeline should be added to improve the clarity of work flow.
Response: Thank you very much for your comments. We have added some contents in figure 3. but due to the different trends and directions of this study, it is difficult for us to change the table into an appropriate picture for display in the results of bioinformatics analysis.

Reviewer Response:
I do not understand why the author is not able to add a graphical work flow how the micro-array data sets were analyzed. These work flow are common in studies re-analyzing micro array data and should be demonstrated to estimate quality of the analyses.
I can´t detect anything in figure 3 which is associated with such anwork flow overview.

4. Reviewer 1: Comment 8.

a. Please edit Line 271-282, this part is quite confusing because lower expression in COPD of TLR2 and IL-10 in other works are described, you demonstrate higher levels but do not discuss the differences.
Response: Thank you for your valuable comments. We have discussed the difference which was highlighted in red in the discussion part.
Line 316-321 (track changed manuscript):
Most of studies showed that the decreased IL-10 level in airways is related to the pathogenesis of chronic airway inflammation in COPD (Takanashi et al. 1999). Lower IL-10 level was reported to be associated with a higher frequency and severity of COPD(Sun et al. 2013b). The present study about IL-10 in COPD was different. We are confusing that the results showed the IL-10 expression level a rising trends from stage I to IV.

Reviewer Response:
Here the improvements of the authors are quite sparse, do you have any speculations why IL-10 is downregulated? For example, what are differences in the methodical setup between the works?


Minor:
1. Please explain how the TLR2 Kit works.
a. How do you detect TLR2 on the surface of monocytes using a realtime PCR (Line 183)
b. Please describe precisely how the differences in expression levels were detected.
2. Information given in the figure legends are sparse and should be extended.
3. Please check orthography, there are still sentences and sections which are hard to understand.

·

Basic reporting

The authors addressed all concerns raised by the reviewers and the manuscript has been significantly improved. It is suitable for publication in PeerJ.

Experimental design

NA

Validity of the findings

NA

Additional comments

NA

---

## Round 0.3 · Major Revisions

Reviewer 1 has raised several critical points about this manuscript. These comments are either information seeking (information like the chosen housekeeping gene) or method ambiguity (PBMC consists of lymphocytes as well as monocytes). Please go through the reviewer 1 comment carefully and try to address them precisely. If needed, perform an experiment to support your claim (Flow cytometry for TLR-2 expression)

Reviewer 1 ·

Basic reporting

Please find response in general comments

Experimental design

Please find response in general comments

Validity of the findings

Please find response in general comments

Additional comments

First of all I want to thank the authors for all the efforts they done to improve the manuscript.
Unfortunately, there are still several deficiencies in the logical structure, experimental setup and language expression. Moreover, the present manuscript revealed unpredicted errors in basic reporting, experimental design and validity of the findings.
The present manuscript is now reasonably comprehensible written; unfortunately there are still many general deficiencies in language expression.
For the first time, method descriptions like TLR-expression analysis are presented, but still information like the chosen housekeeping gene or the calibrator used to perform the delta/deltaCt method are missing. Moreover, in line 319 of the revised manuscript they describe that TLR2 expression seems to be cloth to the detection level (low Ct values and that this might led to experimental errors. There are still some uncertainties, the authors described that they isolated PBMC and analyze the mRNA expression of TLR-2 in the monocytes. However, PBMC consists of lymphocytes as well as monocytes, were the cell types separated?
The chosen method, analyzing TLR-2 expression on mRNA level, seems an inappropriate approach, comparing the expression of a surface receptor of the innate immunity. There could be wide differences between mRNA and protein expression. Here methods, like flow cytometry are recommended analyzing surface expression of the receptors. This analysis is mandatory, especially because experiments of others demonstrate no differences in these levels on blood monocytes of healthy, smokers and COPD patients [for example: Scheele, Respiratory Medicine, 2011].
The authors detected in the second revision that the showed wrong results for IL-10. IL-10 was not a hub gene in the PPI network. They excluded IL-10 from the introduction and the detailed description in the discussion. However, there are still parts in the discussion [line 297], describing IL-10 as a hub protein. Thank you for the disclosure of the wrong presented results. Unfortunately, this misanalysis in the central focus of the manuscript, the evaluation and detection of biomarkers in a preexisting dataset, gives a weak expression of the overall quality in the analysis process.
Moreover, the authors try to evaluate expression levels of CD79A. However, here no differences in their cohort were detectable.
Due to all this points revealed in the revision process and the fact that results and methods are still not clearly presented and improved upon three revision cycles I would recommend to reject the manuscript.

---

## Round 0.4 · accepted · Accept

Based on my assessment of all the questions and Authors response we should accept this manuscript.

Reviewer 1 ·

Basic reporting

Please find response in general comments

Experimental design

Please find response in general comments

Validity of the findings

Please find response in general comments

Additional comments

As I mentioned in my last revision the present manuscript revealed several unpredicted errors in basic reporting, experimental design and validity of the findings during the revision process.
Some points were improved by the authors for example: Language Expression.
Still my concerns of the last revision are not completely eliminated and again new questions are raised.
a. Why are only monocytes analyzed for TLR2 expression and not PBMC completely? TLR2 is also expressed on lymphocytes.
b. The expression of this marker was only on mRNA level, and here expression was cloth to the detection level (low Ct values and that this might led to experimental errors (as mentioned by the authors themselves).
c. The authors add information upon the housekeeping gene, but did not add any information upon the calibrator.
d. The chosen primer seem to be specific for TLR2 transcript variant 2, why only analysis of this variant is chosen is not mentioned.